# Contour Line Stylization to Visualize Multivariate Information

Gazi Md. Hasnat Zahan*        Debajyoti Mondal†        Carl Gutwin‡

Department of Computer Science, University of Saskatchewan
Saskatoon, Saskatchewan, Canada

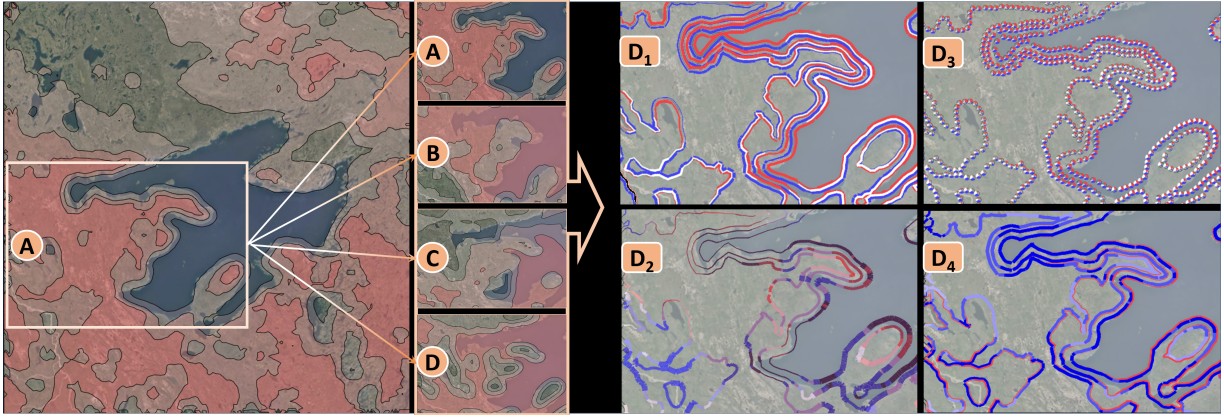

Figure 1: (left) A geographic map, and the contour plots of four climatic parameters A (albedo), B (soil moisture), C (pressure), and D (temperature) on a part of the map. (right) Four of our five designs that encode B, C, and D along the contour lines of A.

## ABSTRACT

Contour plots are widely used in geospatial data visualization as they provide natural interpretation of information across spatial scales. To compare a geospatial attribute against others, contour plots for the base attribute (e.g., elevation) are often overlaid, blended, or examined side by side with other attributes (e.g., temperature or pressure). Such visual inspection is challenging since overlay and color blending both clutter the visualization, and a side-by-side arrangement requires users to mentally integrate the information from different plots. Therefore, these approaches become less efficient as the number of attributes grows.

In this paper we examine the fundamental question of whether the base contour lines, which are already present in the map space, can be leveraged to visualize how other attributes relate to the base attribute. We present five different designs for stylizing contour lines, and investigate their interpretability using three crowdsourced studies. Our first two studies examined how contour width and number of contour intervals affect interpretability, using synthetic datasets where we controlled the underlying data distribution. We then compared the designs in a third study that used both synthetic and real-world meteorological data. Our studies show the effectiveness of stylizing contour lines to enrich the understanding of how different attributes relate to the reference contour plot, reveal trade-offs among design parameters, and provide designers with important insights into the factors that influence interpretability.

**Index Terms:** Human-centered computing—Visualization—Visualization techniques; Human-centered computing—Visualization—Visualization design and evaluation methods

---

*e-mail: gazi.hasnat@usask.ca
†e-mail:d.mondal@usask.ca
‡e-mail:carl.gutwin@usask.ca

## 1 INTRODUCTION

Contour plots are widely used to visualize geospatial information on two-dimensional maps. Contour lines and contour intervals are two important features of a contour plot. A contour line (isoline) represents a fixed threshold value and connects map points having that value. A contour interval corresponds to a range of values within the bounds indicated by two successive threshold values.

The simplicity and rich information found in contour plots make them a popular choice for infographic posters and in geospatial data analysis [5,13,32]. Contour lines provide us with a potentially-useful visualization resource – a set of points that are already on the map. We can leverage these points to show other data attributes along the contour line, which can provide insights into how other geospatial attributes relate to the base attribute. To the best of our knowledge, effectiveness of contour line stylization and the boundaries of human perception to interpret them are not well understood.

In this paper, we examine how to stylize contour lines to provide useful additional information to the viewer (Figure 1). For three variables, it is common to plot the contour lines of one variable and then to blend the filled contour plots of the two other variables [15, 25, 46]. Therefore, we focus on the case of four variables, i.e., we augment the contour lines of one variable with the information from three other variables.

We do not focus on any domain specific application, but rather attempt to improve our understanding of various facets of contour line stylization. The contour plots may result from geospatial datasets, mathematical surfaces, or even scatterplot densities. However, there exist several motivating scenarios (e.g., analyzing historical change in contour lines or understanding correlation) where contour stylization may be useful. Figure 2 shows such a motivating example based on front prediction in meteorological analysis. The development of a front depends on several factors such as temperature, moisture, wind direction and pressure. Figure 2 (left) shows front prediction by the National Oceanic and Atmospheric Administration (NOAA) Weather Prediction Center (WPC) archive, where the curved lines (red, blue or mixed) correspond to various types of fronts (warm, cold or stationary fronts, respectively). Note that such fronts can be

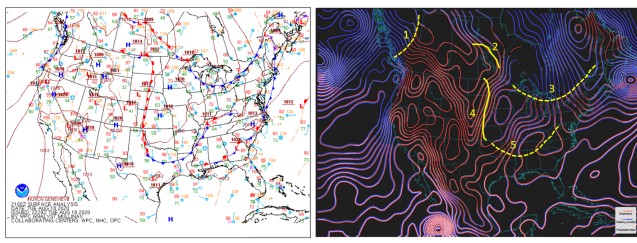

Figure 2: (left) Front detection by NOAA WPC. (right) detection using one of our five visualization techniques.

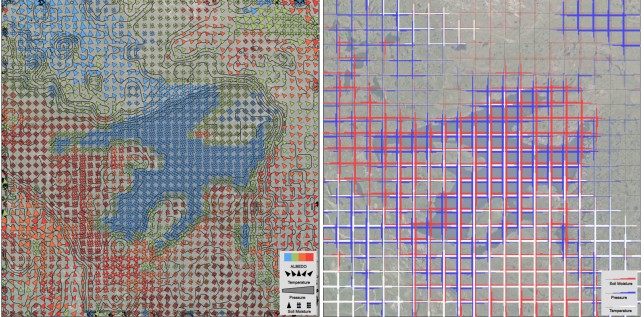

Figure 3: Multivariate visualization with (left) glyphs occludes the map, and (right) grid stylization lacks the gradient information.

derived using software or by painstaking inspection of the numbers plotted on the map representing various weather parameters. Figure 2 (right) shows our contour stylization for 4 weather variables (pressure, temperature, relative humidity and precipitable water). The contour lines represent isobars (pressure). The temperature, relative humidity and precipitable water are encoded in red, blue, and white lines respectively. Contour line stylization can readily reveal some potential cold fronts (yellow curves 1, 3, and 5) and warm fronts (curves 2 and 4), which shows the potential of using contour line stylization alongside traditional visualizations.

Multivariate visualizations that encode data attributes into different preattentive perceptual features of a visual element (glyph) [3, 36, 43] such as size, shape, color, and texture, are typical ways to visualize geospatial information on a map. A well-known limitation of a glyph-based visualization is that it clutters the map [10]. While a dense overlay occludes the view of the base map (Figure 3 (left)), a sparse overlay compromises perception of geospatial connectedness and lacks the gradient information that naturally comes from a contour plot (Figure 3 (right)).

**Our Contribution:** We consider geospatial data with four attributes – A, B, C and D – and encode B, C and D along the contour lines of A. We design five visual encodings and investigate whether users can interpret the attribute values (high, low), trends (increasing or decreasing), and relationships (similar or opposite trends) along a contour line, or across a set of contour lines. Since the encoding position of B, C, and D is determined by A's contour line, users may want to vary the number of contouring thresholds for A, or use a different base contour plot. Therefore, we describe how to design a synthetic dataset to examine the influence of various design parameters through controlled experiments.

We conducted three crowdsourced studies that evaluate our designs. The first two studies reveal how contour width and the number of contour intervals influence the visual interpretability of our designs. The third study used both synthetic and real-world meteorological datasets to assess how the designs and datasets compared in terms of task completion time and accuracy, for common geospatial

data analysis tasks. In addition to revealing insights into our designs, our experimental results also suggest that results obtained using synthetic datasets generalize to real-world datasets.

## 2 RELATED WORK

### 2.1 Multivariate Visualization on a Map

Geospatial data are often shown using choropleth maps [26, 29], and contour plots [28]. While choropleth maps and cartograms [14] reveal properties of a region, a contour plot helps to understand the data distribution on a map and find regions with similar properties. Data analysts often use color blending for finding probable correlations between two geospatial variables [15]. However, creating a high-quality bivariate choropleth or contour map requires careful choice of blending colors and textures [24].

Researchers have also attempted to construct trivariate choropleth maps using the CMY color model [6, 37]. Wu and Zhang [46] examined a 4-variate map that captures the contour band information for each variable in thin visual ribbons, and then overlays the ribbons for all four variables using four different colors. Overlaying glyphs [31, 41] or charts [4, 16] on a map is a popular way to visualize geospatial information. Glyphs are often designed to encode data into features that can be perceived through preattentive visual channels [45]. A rich body of visualization design research examines how humans perceive various combinations of geometric, optical, relational, and semantic channels. We refer readers to recent surveys [10, 17, 43] for a detailed review of glyph design. Glyph-based visualizations often must use a careful glyph positioning technique [31, 46], as creating glyphs for many data points on a map creates overlapping.

Various texture metrics such as contrast, coarseness, periodicity, and directionality [27, 40] have been used to visualize multivariate data. Healey and Enns [20] introduced pexels that encode multi-dimensional datasets into multi-colored perceptual textures with height, density, and regularity properties. Shenas and Interrante [38] showed that color and texture can be combined to meaningfully convey multivariate information with four or more variables on a choropleth map.

### 2.2 Stylization of Lines and Boundaries

Stylized lines naturally appear in the visualization of trajectory data. For example, traffic flow data are often color-coded on road networks as heatmaps [23, 42]. Andrienko et al. [1] extracted characteristic points from car trajectories and aggregated them to create flows between cellular areas to reveal movement patterns in a city. They used stylization to depict various information about the aggregated flows. Huang et al. [23] modeled taxi trajectories using a graph. They stylized the streets based on node centrality and overlaid rose charts to visualize other traffic information. Perin et al. [2, 35] investigated combinations of thickness, monochromatic color scheme, and tick mark frequency on a line to encode time and speed on a two-dimensional line. They observed that encoding speed with a color scheme and time using one of the other two features improved user perception.

Geographic cluster visualization and map generation techniques have also considered line stylization. Christophe et al. [9] proposed a pipeline for generating artistic and cartographic maps that integrates linear stylization, patch-based region filling and vector texture generation. Kim et al. [24] created Bristle Maps that put bristles perpendicular to the linear elements (streets, subway lines) of the map and then encoded multivariate information into the length, density, color, orientation, and transparency of the bristles. Zhang et al. [47] introduced TopoGroups that aggregate spatial data into hierarchical clusters, and show information about geographic clusters on the cluster boundaries. Although TopoGroups summarizes cluster information along the boundary, Zhang et al. noted that users may mistakenly see the visualization as representing local statistics

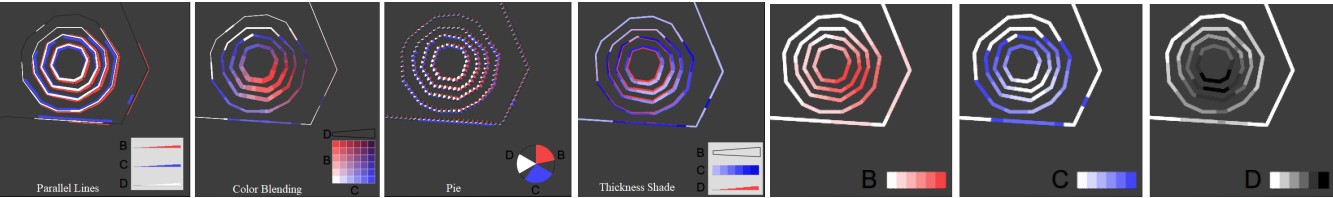

Figure 4: Encoding multivariate information using *Parallel Lines*, *Color Blending*, *Pie*, *Thickness-Shade* and *Side-by-Side*.

near the boundary. In subsequent work, Zhang et al. [48] proposed TopoText that replaces the boundaries using oriented text.

Visual encoding of lines and boundaries has been widely used in visualizing data uncertainty [8, 18]. Cedilnik and Rheingans [8] overlaid a regular grid on the map and then stylized the grid edges using blur, jitter, and wave. Data uncertainty has also been mapped to contour lines, where uncertainty is mapped to line color, thickness, and dash frequency [33]. Line stylization has also been used in cartograms. Görtler [18] proposed bubble treemap that represents uncertainty information using wavy circle boundaries, and varied wave frequency and amplitude based on the uncertainty. Patterson and Lodha [34] encoded five socio-economic variables simultaneously on a world map using country fill color, glyph fill color, glyph size, country boundary color, and cartogram distortion.

## 3 VISUAL ENCODING

In this section we describe five contour-based designs (Figures 4) for encoding geospatial information with four-attributes: A, B, C, and D. We assume that all the attributes are numeric and positive. We create a set of contour lines using the A, and then encode the attributes B, C, and D along the contour lines of A using visual features.

**Rationale:** For encoding, we used visual features that are preattentive [7, 44] and intuitive to interpret, or have been used in prior research [35], e.g., line thickness, monochromatic color scheme, and pie slice. Most of our designs are based on the notion of channel separability [21], but we also kept color blending as it has often been used in the context for correlation analysis in geospatial data [15, 22, 39].

**Design 1 (*Parallel Lines*):** This design maps B, C, and D into three lines with distinct colors. The lines for B and C lie on opposite sides of the contour line of A, and the line for D follows the contour line of A. The data values are encoded using line width (between 0 and $w$), and the value of the attribute is linearly mapped to the range $[0, w]$. If D's value is 0, the base contour line A becomes visible.

**Design 2 (*Color Blending*):** This design encodes B and C with distinct colors, and then blends them on the contour line of A. The attribute D is mapped to the width of the contour line. Note that since the contour line of A has a non-zero width $u$, the values of D are mapped to the linewidth range $[u, w]$. Consequently, B and C remain visible even when D is 0.

**Design 3 (*Pie*):** This design encodes B, C, and D using pie slices of distinct colors, and puts them together to create a pie icon. The only difference from a pie chart is that the sum of the values of B, C, and D may not be equal to the total pie area. The pie icons are placed successively along the contour line of A. The pie slices for B, C, D start at $0°$, $120°$ and $240°$ (assuming the top as $0°$), and can grow clockwise to cover an angle of $120°$. An attribute value is encoded into the angle covered by the corresponding pie slice.

**Design 4 (*Thickness-Shade*):** This design represents B and D using two distinct lines. The lines of B and D lie on opposite sides of the contour lines of A, with values encoded using line width. The values of C are encoded using a monochromatic color scheme, where the color appears on B's line. A low C value corresponds to a lighter shade, and a high value to a darker shade. The minimum line

width for B is set to a positive threshold $u$, making a range of $[u, w]$ so that C remains visible even when B is 0.

**Design 5 (*Side-by-Side*):** This design shows B, C, and D in separate side-by-side views. Each of B, C, and D is encoded using a distinct monochromatic color scheme. The color appears on the contour lines of A. We ensured that the width and height of each of the *Side-by-Side* views to be $\lceil \sqrt{A/3} \rceil$, where $A$ is the total pixel area of any other design, assuming a square display.

## 4 IMPLEMENTATION DETAILS AND DATASETS

The choice for contouring thresholds are application specific. But in our controlled experiment, we used $k$-quantiles as the thresholds. This allows us to reduce visual clutter and to examine the design across a large number of contouring thresholds. We first computed the contour lines for A, and then further processed these polylines by dividing long line segments uniformly to create fine-grained polygonal chains. We then encoded the attributes by interpolating the values at the endpoints of these tiny segments. Figure 5 illustrates the parameters used for the design. Here $b$, $c$ and $d$ denote the normalized B, C, and D values, respectively, and $t$ is a thickness factor that was used to linearly map the attribute values to the input line-thickness range. The number of discrete shades in the perceptual color scale depends on the number of contour intervals. For blending, we used CSS *mix-blend-mode*, where the scheme *Multiply* was chosen in a pilot study comparing 3 possible candidate schemes: *Multiply*, *Darken* and *Difference*.

**Synthetic Data:** For each of the four attributes, we created scatterplots consisting of four Gaussian clusters that were positioned randomly in the four quadrants. Each cluster had 40000 samples, with randomly varying covariance (2.5-7.5), 2 features (x and y coordinate), and 4/6/8 classes (for 4, 6, and 8 contour intervals). The clusters were then interpolated and reshaped such that the point density plot for each cluster takes the shape of a peak or valley.

All the clusters of A, B, C, and D at a quadrant overlapped one another making various peak-valley combinations. This also allowed us to obtain scenarios where an attribute value increases or decreases across successive contour lines of A. To visualize all possible trends (increasing or decreasing) of B, C, and D, we used all possible peak-valley combinations for these attributes. To imitate real-world topographic map patterns, we varied the cluster overlaps for A.

**Real-World Meteorological Data:** To test the designs on real-world data, we used real meteorological datasets [30]. For our study, we extracted 4 attributes from the dataset: *Temperature*, *Pressure*, *Soil Moisture* and *Albedo* from different geolocations.

**Evaluation:** A careful choice for various design parameters is important to achieve optimal readability for the designs. Two major factors that can influence the designs are width (the space allocated along the base contour line for encoding B, C, and D), and the number of contouring thresholds for A. Therefore, we first conducted two studies to examine these factors and choose appropriate parameter values for the designs. In the final study, we evaluated the designs based on viewers' task performance. The first two studies were conducted on synthetic data and the final study included both synthetic and real-world data.

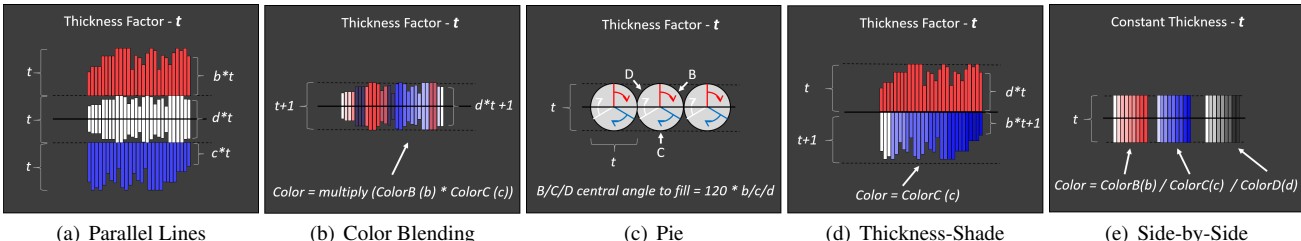

| (a) Parallel Lines | (b) Color Blending | (c) Pie | (d) Thickness-Shade | (e) Side-by-Side |

Figure 5: Illustration for the implementation details for different designs.

Table 1: Different widths for Study 1

| Design | Width Alternatives | | |
| --- | --- | --- | --- |
| | Minimum | Median | Maximum |
| *Parallel Lines* | 9 | - | 12 |
| *Color Blending* | 4 | 7 | 10 |
| *Pie* | 8 | - | 10 |
| *Thickness-Shade* | 8 | - | 10 |
| *Side-by-Side* | 1 | 2 | 4 |

Table 2: Tasks with domains for Study 1

| ID | Task | Domain |
| --- | --- | --- |
| 1 | Select the marked contour region that best represents the following combination: high B, low C, and high D | Compare different marked contour regions |
| 2 | Consider the contour regions intersected by the lines. Select the directed line that best represents the following trend: B and D both increase, and C decreases | Interpret trends across contour lines |
| 3 | Select the marked contour region that, when moving clockwise, best represents the following trend: B decreases, D increases, and C stays the same | Interpret trends along a contour line |
| 4 | Count the number of lines that show the following: B and C have the opposite trend to D | Identify similar/ opposite trends across contour lines |

## 5 STUDY 1 (CONTOUR WIDTH)

Our first study investigated the effect of contour width on design interpretability, as well as on tasks that use the underlying map.

Intuitively, increasing contour width should make the encoded variables easier to see and interpret, but will also increase occlusion of the map; in addition, wide contours may also overlap each other, depending on the density and shape of the contours. To investigate this trade-off, we set the number of contour intervals to 8 (i.e., 7 contouring thresholds), and then determined a range of widths to explore for each design. We used 8 contour intervals because this allows designers a reasonable spectrum of design choices, and gives us a reasonable range to investigate in Study 2 (described below).

Table 1 illustrates the width ranges used in Study 1. We chose minimum and maximum widths for each design based on informal testing with each design's encoding. The minimum width is determined by the number of pixels required to create the design and make the variation in the attributes noticeable. For example, *Parallel Lines* requires 9 pixels to encode the variation (low, mid, high) for each of the three attributes. The maximum width corresponds to the case when the successive linear elements are about to overlap.

We also selected a median width if there was enough difference between minimum and maximum that the median would have at least two pixel units from the extremes. Hence we only have the median width for *Color Blending* and *Side-by-Side* (e.g., for *Parallel Lines*, the encoding for B, C and D needs to be the same, so the next possible width choice after 9 is 12).

### 5.1 S1: Participants, Data, and Tasks

We ran a crowdsourced study on Amazon Mechanical Turk (AMT) [12]. To be eligible, a participant needed to pass a color perception test (Ishihara test [11]), run the study on a desktop computer, reside in North America, and have at least an 80% approval rate in AMT. We recorded 63 complete responses (32 male, 31 female, majority in age range 30-39).

All experimental tasks were created using synthetic data. We tested the 5 designs described earlier, with the width choices determined for that design (12 in total, see Table 1). Participants completed 57 tasks (4 tasks for each width that involved interpreting the variables encoded in the design, plus one additional task for three

of the widths that involved reading the background map).

**Rationale for Tasks:** We chose the tasks to be general enough so that they can be applied in a variety of scenarios, and by considering use cases illustrated in Section 1. We deliberately designed 3-variable tasks since our knowledge of line stylization is most limited in this case. In our four interpretation tasks (Table 2), one involved identifying values, two involved looking for trends, and one involved comparing trends (e.g., Figure 2). For tasks 1 and 3, we marked four contour sections on the design (Figure 6 (left)), and participants selected the option that matched the requested combination or trend. For tasks 2 and 4, we drew four lines across the contours (e.g., see the black arrows in Figure 6 (right)), and participants selected the option that best identified a specific trend.

The background map reading task used the *Color Blending* design with 3 of the widths (4, 8, and 12). In this task, icons were placed in the underlying map, and participants were asked to count the number of icons and select an answer from 4 options. The reason for choosing *Color Blending* as a representative design for the background task is that it has three different width choices that are substantially different from each other. The icons were $8 \times 8$ pixels. The number of icons ranged between 18 to 22, and were placed randomly on the map. Therefore, in some cases the icons were partially hidden by the contour lines.

*S1 Hypotheses:* We hypothesized that as width increases, accuracy will increase and completion time will decrease ($h_1$). We also hypothesized that the influence of width will be more noticeable for *Parallel Lines*, *Pie*, and *Side-by-Side* than for other designs ($h_2$) (because *Color Blending* and *Thickness-Shade* could be more difficult to interpret for inexperienced users). Finally, we hypothesized that for the background task, increasing contour width will lead to lower accuracy and higher completion time ($h_3$).

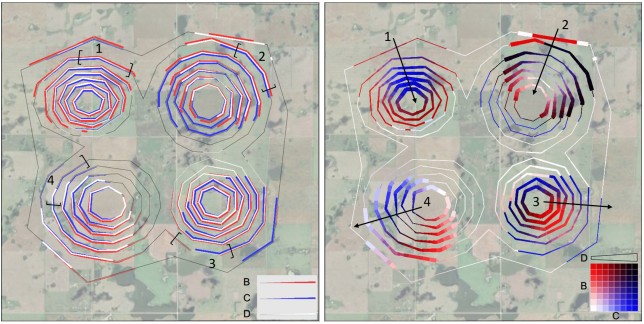

Figure 6: Contour-region task (left); Trend task (right)

## 5.2 S1: Procedure

Participants completed an informed consent form and then were shown a description of the designs and given a set of practice tasks to complete. After each practice task, the participant was told whether the response was correct or not, and were given an explanation of the correct answer with a brief justification. Participants then completed the 57 tasks as described above. After each design, participants completed a NASA-TLX-style effort questionnaire [19] and at the end of the study, they rated their familiarity with visualization interfaces and their preferences for each of the widths. Participants were asked to complete the tasks as quickly and accurately as possible. Each task started with a 'start' button, and ended when the participant selected one of the multiple-choice answer options and pressed 'next'. Before starting a new task, participants were shown a reminder that they could rest before continuing.

The study used a within-participants design, with contour width as the independent variable (considered separately for each design); dependent variables were accuracy, completion time, and subjective effort scores. Designs and tasks were presented in random order (sampling without replacement).

## 5.3 S1: Results

We applied additional filters to test whether participants were legitimately attempting the tasks (e.g., answering inconsistently and large time gaps in their surveys). After filtering, we had 44 participants (20 male, 24 female, majority in age range 31-39).

**S1: Interpretation tasks:** For the four interpretation tasks involving the B, C, and D attributes, data were analyzed using repeated-measures ANOVAs for each design (because each design used a different set of widths); Bonferroni corrected paired t-tests were used for follow-up comparisons. Figure 7 (left) shows that accuracies increased slightly as contour width increased, and Figure 7 (middle) shows that completion times decreased overall as width increased.

We found significant effects of width on accuracy for the *Parallel Lines* design, and on completion time for *Color Blending*, *Pie*, and *Side-by-Side*. No effect of width was found for *Thickness-Shade*. For *Parallel Lines* (using widths 9 and 12), we found a significant effect of width on accuracy ($F_{1, 43} = 5.4$, $p <.05$), with width 12 having higher accuracy than width 9. For *Color Blending* (widths 4, 7, 10), we found a significant effect of width on completion time ($F_{2, 86} = 8.09$, $p <.05$); Figure 7 (middle). Post-hoc t-tests showed that width 10 was faster than both width 7 and width 4 (all $p <.05$). For *Pie* (widths 8 and 10), we found a significant effect of width on completion time ($F_{1, 43} = 9.46$, $p <.05$). Width 10 was faster than width 8. For *Side-by-Side* (widths 1, 2, 4), we found a significant effect of width on completion time ($F_{2, 86} = 13.46$, $p <.05$). Post-hoc t-tests showed widths 2 and 4 to be faster than width 1 ($p <.05$).

**S1: Background Task:** The background icon-counting task used design *Color Blending* with widths 4, 8, and 12. We found a significant effect of width on accuracy ($F_{2, 86} = 121.76$, $p <.05$). Post-hoc

t-tests showed significant differences among all 3 widths ($p <.05$). As shown in Figure 7 (right), the mean accuracy for width 4 was higher than 8, which was higher than that of 12. There was no effect of width on completion time.

**S1: Effort and Preference:** We asked participants to rate their amount of mental effort, overall effort, frustration, and perceived success with each design. Friedman tests showed significant differences in all questions (all $p <.005$), with *Parallel Lines* and *Side-by-Side* rated better than the other designs. The width preference question reveals higher user preferences for the maximum (50% of participants) and median (41%) widths.

## 5.4 S1: Discussion

Increased contour widths for interpretation tasks led to improved completion time in three of the designs, and improved accuracy for one design, partially supporting hypothesis $h_1$. We did not find any significant effect of width for *Thickness-Shade*, which partially supports our hypothesis $h_2$ that suggested the effects of width would be more obvious for some designs. Our results for the background task (effect of width on accuracy, but not on completion time) partially support hypothesis $h_3$. Overall, the fact that there was only a minor effect of reduced width on interpretability (particularly for accuracy) means that width can often be safely reduced in scenarios where the visibility of the background is critical.

Based on these findings we chose to use the maximum width for each design in further studies. In the following section, we explore the influence of contour intervals, which is another important element of a contour plot.

## 6 STUDY 2 (CONTOUR INTERVALS)

A larger number of contour intervals increases both the number of visual elements in the design and the degree of background occlusion. Increasing the number of contours, however, also provides more data points for the other variables visualized on the contour, and so may increase the interpretability of these variables. Our study explores this trade-off using a study design similar to that used above.

### 6.1 S2: Participants, Data, and Tasks

We ran the study on Amazon Mechanical Turk with the same eligibility criteria as in Study 1. We recorded 68 complete responses (40 male, 26 female, 1 non-binary, 1 preferred not to answer), aged 21-60 (majority in age range 21-29). None of the participants took part in Study 1. The study used the 5 designs described above, each with 3 contour interval alternatives (4, 6, or 8 contours). Participants completed 60 tasks: the same 4 interpretation tasks from Study 1 for each combination of design and contour interval, and the background icon-counting task.

For analysing the interpretation tasks, the study used a within-participants design with three factors: Design (the five designs described above), Task (the four interpretation tasks from Study 1), and Number of Intervals (4, 6, or 8). The main dependent measures were accuracy and completion time; we also collected subjective effort and preference scores.

*S2 Hypotheses:* We hypothesized that more contour levels will result in better performance for the tasks that require analysis across contour lines ($h_4$). For the background task, we hypothesized that more contour levels will lead to lower accuracy and higher completion time ($h_5$), due to increased occlusion.

### 6.2 S2: Procedure

Similar to Study 1, participants went through the eligibility tests, design demonstration, and practice tasks. They then completed the main study tasks and filled out the TLX-style effort surveys and overall preference questions. The data and tasks were the same as in Study 1, and designs and tasks were presented in random order.

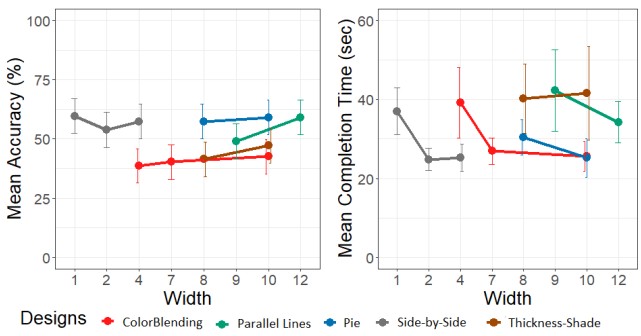
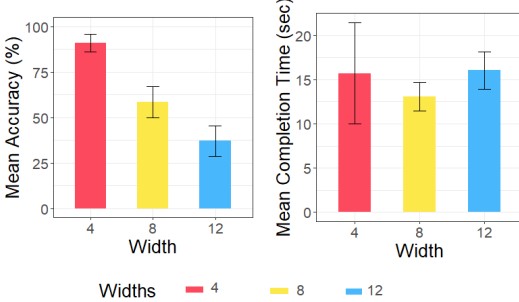

Figure 7: Study 1: (left and middle) Performances of the designs at different width choices for Tasks 1-4. Lines are connected to group the designs, but not to denote continuity of the width. (right) Accuracy and completion time for the icon-counting task.

## 6.3 S2: Results

After filtering the participants based on response consistency, we had 46 participants (28 male, 1 non-binary, 17 female, majority in age range 31-39).

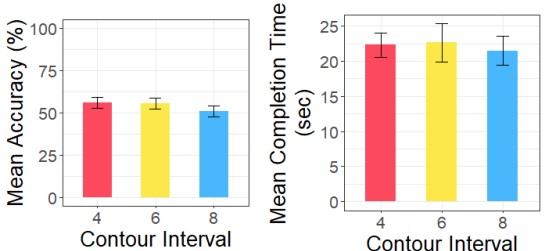

Figure 8: Study 2: Task performance with different contour intervals.

**S2: Interpretation tasks:** We carried out $5 \times 4 \times 3$ RM-ANOVAs (Design $\times$ Task $\times$ Number of Intervals) for both accuracy and completion time, with Bonferroni-corrected t-tests as follow-up. There was a significant main effect of Intervals ($F_{2, 90} = 3.69$, $p <.05$) on accuracy. Post-hoc tests showed that 4 and 6 contour intervals had higher accuracy than 8 intervals (see Figure 8). There was also an interaction between Design and Task ($F_{12, 540} = 2.09$, $p <.05$). As can be seen in Figure 9, the *Pie* design had higher accuracy in Tasks 2 and 3 compared to the other designs. There was no main effect of Design on accuracy ($F_{4, 180} = 1.58$, $p =0.18$). For completion time, there were no main effects of Design or Intervals ($p >.05$), but there was an interaction between Intervals and Task ($F_{6, 270} = 3.21$, $p <.05$).

**S2: Background Task:** We found no main effects of the number of contour intervals on either completion time or accuracy for the icon-counting task.

**S2: Subjective Effort and Preferences:** Participants rated mental effort, overall effort, frustration, and perceived success with each design. Responses were similar across all designs, and Friedman tests showed a significant difference only for overall effort ($p <.05$), with the *Pie* design seen as requiring more effort than the others. We asked participants about their preference: 4 intervals (36%) and 6 intervals (39%) were preferred to 8 (25%).

## 6.4 S2: Discussion

Overall, 4 and 6 intervals performed better than 8. One main reason for this result is that fewer contour intervals result in less visual clutter. Although there was a significant effect of contour levels on accuracy only for Task 3, there were significant interactions between contour intervals and task for completion time, which partially supports $h_4$. We observed that higher contour levels slightly reduced

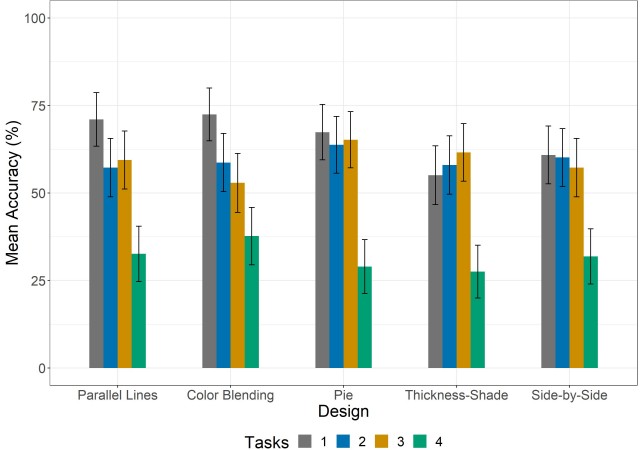

Figure 9: Study 2: Task accuracy for the five designs.

mean accuracy for Tasks 1 and 3, where users may have found it difficult to follow along a line where there were other parallel lines in close proximity.

The interactions show that task performance is dependent on design, contour intervals, and task combinations. The significant interaction (for accuracy) between contour levels and tasks suggests that the association between contour level and designs depends on tasks. Similarly, for completion time, the association between design and contour level depends on tasks.

In the next study, we focus on how performance varies by design in both synthetic and real-world datasets. We used 4 contour intervals for all the designs.

## 7 STUDY 3 (DESIGN COMPARISON)

### 7.1 S3: Participants, Data, and Tasks

We ran the study on Amazon Mechanical Turk with the same eligibility criteria as in Study 1. We recorded 78 complete responses (41 male, 33 female, 2 non-binary, 1 preferred not to answer, majority in age range 30-39). None of them participated in Study 1 or 2.

We used two datasets for this study (one synthetic and one from a real-world scenario). Participants completed 6 different tasks for each design and dataset combination, which resulted in 60 tasks. Of the 6 tasks, 4 were similar to studies 1 and 2. We added 2 additional tasks (Table 3) to examine whether users are able to interpret the extent of value changes of a single attribute (Task 5), and to estimate the difference between two attributes (Task 6).

Similar to Study 1, participants went through the eligibility tests,

Table 3: Tasks with domains for Study 3

| ID | Task | Domain |
|---|---|---|
| 5 | Select the marked contour region that has the maximum change in (a given attribute) | Estimate value changes along a contour line |
| 6 | Select the marked contour region that has the minimum difference between (a given pair of attributes) | Estimate value difference along a contour line |

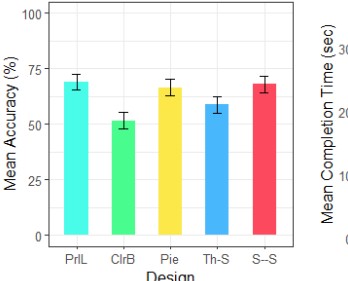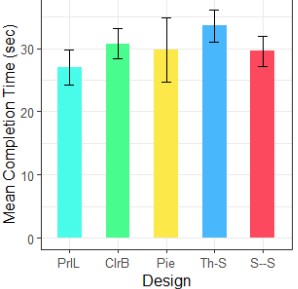

Figure 10: Study 3: Overall performance of the different designs.

Table 4: Study 3: Design Preference Survey on a 0-4 Scale (0: not preferred to 4: highly preferred)

| Design | Average Preference Scores | | | | | | |
|---|---|---|---|---|---|---|---|
| | Task 1 | Task 2 | Task 3 | Task 4 | Task 5 | Task 6 | Overall |
| *Parallel Lines* | 2.28 | 2.5 | 2.48 | **2.37** | 2.57 | **2.74** | **2.85** |
| *Color Blending* | 1.85 | 1.83 | 1.87 | 1.57 | 1.76 | 1.93 | 1.93 |
| *Pie* | 1.78 | 1.87 | 2.24 | 1.72 | 2.06 | 2.02 | 2.15 |
| *Thickness-Shade* | 2.04 | 2.09 | 2.09 | 1.87 | 2.23 | 2.03 | 2.17 |
| *Side-by-Side* | **2.69** | **2.74** | **2.85** | **2.37** | **2.69** | 2.59 | **2.93** |

design demonstrations, and practice tasks. They then completed the main study, the effort survey, and preference questions (in this study, participants stated their preference for one of the designs after each task, as well as overall).

## 7.2 S3: Procedure

The study used a within-participants design with three factors: Design (the five designs described above), Task (the six interpretation tasks), and Dataset (Synthetic or Real-world). The main dependent measures were accuracy and completion time; we also collected subjective effort and preference scores. Designs and tasks were presented in random order (sampling without replacement).

## 7.3 S3: Results

After filtering based on response consistency, we had 54 participants (30 male, 22 female, 2 preferred not to answer, majority in age range 31-39); 45 participants reported familiarity with data visualization interfaces.

We carried out $5 \times 6 \times 2$ RM-ANOVAs (Design $\times$ Task $\times$ Dataset) for both accuracy and completion time, with Bonferroni-corrected t-tests as follow-up. There was a significant main effect of Design ($F_{4, 212} = 19.2$, $p < .001$) on accuracy. Post-hoc t-tests showed that *Parallel Lines*, *Pie*, and *Side-by-Side* were significantly more accurate than *Color Blending* and *Thickness-Shade* (Figure 10). There was also a main effect of Dataset ($F_{1, 53} = 13.8$, $p < .001$): participants were more accurate with the real-world dataset (66%) than with the synthetic dataset (59%).

There were also significant interactions between Design and other factors. First, there was a Design $\times$ Dataset interaction ($F_{4, 212} = 10.7$, $p < .001$). As shown in Figure 11, the *Color Blending* design had substantially lower accuracy for the synthetic data compared to the other designs, and only *Parallel Lines* was equally accurate with both datasets. Second, there was a Design $\times$ Task interaction ($F_{20, 1060} = 5.01$, $p < .001$). Figure 11 shows substantial differences in the tasks depending on the design: for example, the accuracy of the *Color Blending* design was substantially lower in Tasks 1 and 6, and the accuracy of *Thickness-Shade* was lower for Task 6.

For completion time, there were no main effects of Design ($p > .05$), but there were interactions between Design and Dataset ($F_{4, 212} = 2.91$, $p < .005$) and between Design and Task ($F_{20, 1060} = 2.17$, $p < .001$).

**S3: Subjective Effort and Preferences:** We again asked participants to rate mental effort, overall effort, frustration, and perceived success with each design. Friedman tests showed significant differences for all questions ($p < .05$), with *Parallel Lines* and *Side-by-Side* scoring better than the other designs. We also asked users to rate the designs on a 0-4 scale (0: 'not preferred' to 4: 'highly preferred'). Participants rated the designs for each task as well as their overall preference. Mean participant scores are presented in Table 4 (higher values are better). The table highlights the top scores for each task and the top two designs for overall preference (*Parallel Lines* and *Side-by-Side* were the most-preferred designs).

## 7.4 Overall Discussion

The main finding of the third study is that all of the designs were successful for at least some of the tasks, and that the designs were similar in their performance – with the exception of *Color Blending*, which showed reduced accuracy compared to the other designs for Tasks 1 and 6. The study also clearly showed that integrating multiple variables into a single contour line results in visualizations that users can interpret successfully – as successfully as separate individual presentations (*Side-by-Side*). This is an important result for situations where designers need to provide a single larger view rather than divide the available space into pieces, as is needed for a side-by-side presentation.

Overall, two designs–*Parallel Lines* and *Side-by-Side*–performed best and had high preference scores. Both designs have separate encoding space for all the variables; in addition, the encoding for different attributes was similar, and they are intuitive to read without any close inspection of the legend. *Side-by-Side* had high preference scores for all tasks except Task 6: in this task, a higher mean preference for *Parallel Lines* is likely due to its symmetric encoding for all the attributes, which makes the value difference easier to estimate, whereas for *Side-by-Side* users need to compute the difference inspecting two separate views.

Interestingly, however, both the *Parallel Lines* and *Side-by-Side* designs have space constraints (*Parallel Lines* in terms of contour width, and *Side-by-Side* in terms of display space). For scenarios where a single view is required and the visibility of the background is important, neither of these designs may be feasible. In these cases, the *Pie* design appears to be a reasonable compromise because it had good accuracy and takes less space.

The *Color Blending* and *Thickness-Shade* designs both had poor performance on at least one task. This could be due to participants' unfamiliarity with the encoding, but the visual variables used in these designs may be more difficult to interpret overall. In addition, the encoding in these two designs was not symmetric compared to the

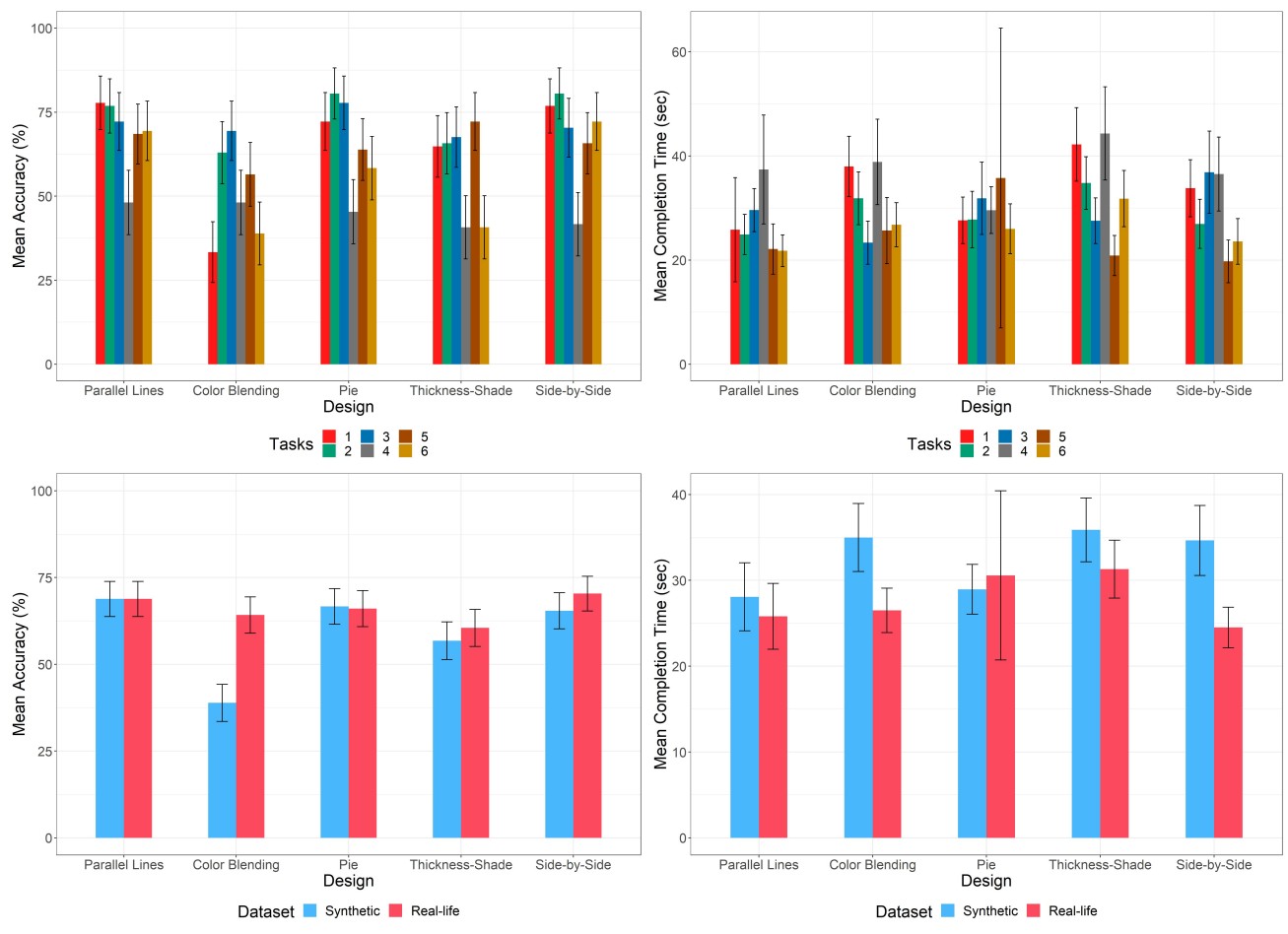

Figure 11: Task performances in Study3, for (top) different designs, and (bottom) different datasets.

other designs. Problems in Task 1 (for *Color Blending*) may have resulted from the need to estimate attribute value combinations, where the interpretation of the designs likely demanded a close inspection of the legend. In addition to the possibility of misinterpretation, this may have led to increased cognitive load or reduced effort for tasks using these designs. The same reasoning holds for the poor performance of *Thickness-Shade* for Task 6, where estimating the value difference between a pair of attributes encoded in different features such as thickness and color shade requires a careful reading of the legend.

Our results showed better performance with the real-world data than with the synthetic dataset (Figure 11), which may be due to differences in the underlying data distributions. The synthetic dataset consisted of almost all possible trend combinations for various attributes, so the corresponding visualizations consisted of highly varied feature combinations; in contrast, the visualizations generated from real-world data had fewer variations. It is likely that visualizations with real-world datasets are visually simpler, leading to improved accuracy and task completion time. These differences partially explain the significant interactions among design, dataset, and task combinations in accuracy, and interaction between design and dataset in completion time.

Based on the study results, we formulated a table of design recommendations (Table 5) that summarizes the preferred design choices for various tasks over three environments—general use, time-sensitive interpretation, and high accuracy requirements. The table shows some strengths for particular designs that are different

from the overall discussion above: for example, if the task requires quick estimation for trends along a contour line, then *Color Blending* and *Thickness-Shade* may be the best design options.

## 8 LIMITATIONS AND FUTURE WORK

Our experience with the designs indicate some limitations in the research that provide opportunities for additional study. First, since we encode the variables at the pixel level, our current implementation does not scale well with large maps; however, rendering techniques using GPU acceleration can be used to overcome this obstacle. Second, as the number of contour intervals grows, the contour lines may sometimes overlap. Therefore, finding an adaptive choice of contouring thresholds or allowing users to interactively choose the base attribute can be a valuable avenue of future research. Third, using existing contours as the basis for additional variables is limited by the density of these contours. Further work is needed on how to represent variables in areas with few contours: for example, adaptive sampling could be used to achieve minimum contour density across the map, or glyph-based techniques could be used to show important changes that occur between contours. Fourth, since we used colors in many of our designs, the interpretability of the designs could depend on the background map colour and texture. Therefore, real-world deployments of our technique will benefit from methods that tune color choices to the background map, or by implementing controls that let users choose the opacity of the background map.

In addition, while the crowdsourced surveys have been found to be useful in gaining insights about our designs, additional controlled

Table 5: Design Recommendation Table

| Domain | Time Sensitive | Environment Accuracy Sensitive | General |
|---|---|---|---|
| Compare different contour parts | *Parallel Lines*, *Pie* | All except *Color Blending* | All except *Color Blending* |
| Search for a trend across contour lines | *Parallel Lines,Pie,Side-by-Side* | *Pie,Side-by-Side* | *Parallel Lines,Side-by-Side,Pie* |
| Search for a trend along a contour line | *Color Blending,Thickness-Shade* | All | All |
| Identify rate of change of a variable along a contour line | All except *Pie* | All | All |
| Identify the value difference on a contour part | *Parallel Lines,Side-by-Side* | *Parallel Lines,Side-by-Side* | *Parallel Lines,Side-by-Side,Pie* |

studies in the lab as well as focus groups with meteorological experts could provide more information. For example, the use of eye tracking for our approach would give more detail on how users interact visually with the different designs. Given the complex interaction among various factors that we observed, in-depth observation of the use of the designs in realistic tasks could help better understand some of the effects and interactions. Finally, we plan to apply our designs to different real-world datasets and explore different contour-based tasks and scenarios that can be used in real geospatial settings.

Our results showed better performance with real-world data than with the synthetic data. It would thus be interesting to explore ways for designing synthetic data that mimics real-life scenarios, e.g., sampling multivariate functions where the variables already have some explicit relations. This may better reveal participants' ability to analyze the data from a given contour line stylization.

## 9 CONCLUSION

Contour plots are widely used, but standard techniques for adding multivariate visualizations onto these plots can clutter the display. To address this problem, we explored how contour lines on a geospatial map can be stylized to encode other attributes in the data. Such a multivariate representation can reduce visual clutter by leveraging the existing contour line space. We designed five types of visual encoding, and examined how various contour parameters such as width and contour levels influence task performances. Our crowdsourced study results showed that participants were able to perform several types of multivariate data analysis tasks with reasonable accuracy, which reveals the potential of our approach.

## ACKNOWLEDGMENTS

We thank all the participants for their participation and feedback. This work is supported by the Natural Sciences and Engineering Research Council of Canada (NSERC), and by two CFREF grants coordinated by GIFS and GIWS.

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
