# OpenReview forum: "Contour Line Stylization to Visualize Multivariate Information"
_graphicsinterface.org/Graphics_Interface/2021/Conference/Second_Cycle — GI 2021_

### Official Review · Reviewer_bd3c · 2021-04-17
**Excellent work**

**Rating:** 8
**Confidence:** 3

**Review:**

This paper presents a set of designs for visualizing multivariate information on contour lines and validates the designs with a set of three user studies. Overall the paper is strong, well-motivated, and well-structured.

One confusing part is regarding the four attributes the techniques are encoding, which are labelled as A, B, C, and D, without any real physical meaning. I suggest the authors give more concrete examples about the attributes to encode. It is very difficult to understand in the introduction and later in Sec 3. Also, why four attributes? Why not five? The number sounds a bit arbitrary and it needs further justification.

Furthermore, it is unclear how the characteristics of the attributes would affect the design and encoding. For example, are the four attributes numerical, ordinal, or categorical? The authors seem to assume numerical attributes. Some discussion around this line should be included, because if the variables are in different types, the visual designs should change accordingly. I suggest the authors refer to Capendale's visual variables [1].

The user studies are well designed and the tasks are reasonable. However, all the figures are blurry. High-resolution figures are required for high-quality digital publications.

[1] Sheelagh Carpendale. 2008. Considering Visual Variables as a Basis for Information Visualisation.

---

### Official Review · Reviewer_JQfW · 2021-04-17
**The paper presents a user study on different ways of multiple encodings of contour lines. The study is well performed and the results are along the lines of the expected with some twists.**

**Rating:** 9
**Confidence:** 4

**Review:**

The paper explores the design space of contour lines, especially when multiple variables (up to 4) are being visually encoded at the same time. Through multiple amazon turk studies, different insights are produced as to how different contour frequencies and contour thickness impact the interpretability of these contour stylings.

The paper is well written. The results are confirming obvious aspects (such that side-by-side encodings are most accurate) and also reveal some surprising aspects (such that glyphs seem to work well). Overall, I very much favor the publication of the paper. However, I would appreciate if some minor issues will be addressed.

* "or has been used in prior" should be "or have been used in prior"
* how come that the median age is a range and not an age?
* "drew four lines across the contours": this concept was not clear to me.
* Fig 7: spell out the designs for the legend
* Fig 7, right: color coding unclear
* Table 4: explain the numbers in the caption (range+pref)

references:
  - many of the references have problems with capitalizations. Many of the journals / conf names appear in lower case.
  - ref 9+10 is duplicated
  - ref 8, 34 have "et al.": please list all authors
  - ref 4, 11, 12, 19, 23, 30 have a wrong capitalization of the title
  - ref 22 is missing details

---

### Official Review · Reviewer_mQCg · 2021-05-03
**Solid, well thought out study providing useful design insights**

**Rating:** 8
**Confidence:** 3

**Review:**

The paper explores how contour-lines can be used to effectively visualize co-variation of multi-variate data over a spatial domain. A focal point of the study is the simultaneous visualization of 4 variable using one of 4 different encodings. Several studies are performed using amazon turk to evaluate the proposed encodings. These studies are well-thought out and nicely reported, providing useful insights into the effectiveness of each encoding in different scenarios. The author’s findings are synthesised into a practical set of design recommendations (Table 5).

Overall, the paper appears to be excellent work. There are a few minor points I wonder about:
1)	As the synthetic data is randomly generated, users are in essence asked to interpret spurious correlations in the data. I wonder if using data where there are explicit relations between variables (e.g. B is inversely proportional to C) may provide a better test of participants ability to reason about data for a given visualization.
2)	Why 4 variables (A, B, C, D)? This could be better justified.

One aspect of design that’s seems important for further study, is how to decide on the mapping of variables to different aspects of the visualization in asymmetric encodings (e.g. for Thickness-Shade, which variable is used to generate contours vs line-width vs color).

---

### Meta-Review · Area_Chair_Tu2D · 2021-05-05

**Recommendation:** Accept
**Confidence:** 4

**Metareview:**

The paper presents and evaluates designs for the visualization of multivariate data using contour lines. The paper is well-written, and the four user-studies performed to evaluate visualizations (using amazon turk) are well-designed. The studies yield a combination of expected and surprizing results, and are nicely synthesized into a practical set of design recommendations. Reviewers also raise several issues that, if addressed, would strengthen the paper. The choice of multivariate data used in the study (4 randomly generated numerical attributes) could be better justified or augmented with real-world examples. It would also be helpful to comment on how the characteristics of attributes affect design and encoding. Reviewer 2 provides a list of minor issues to be addressed in the final version, which should include high-resolution figures. In summary, as recognized by all reviewers, this is excellent work that warrants acceptance at GI.

---

### Decision · Program_Chairs · 2021-05-08

Accept